# Heterologous expression and characterization of mutant cellulase from indigenous strain of Aspergillus niger

**Waqas Ahmad, Muddassar Zafar** *, **Zahid Anwar**

Department of Biochemistry and Biotechnology, University of Gujrat, Hafiz Hayat Campus, Gujrat, Pakistan

* muddassar.zafar@uog.edu.pk

## Abstract

The purpose of current research work was to investigate the effect of mutagenesis on endo-glucanase B activity of indigenous strain of *Aspergillus niger* and its heterologous expression studies in the *pET28a+* vector. The physical and chemical mutagens were employed to incorporate mutations in *A. niger*. For determination of mutations, mRNA was isolated followed by cDNA synthesis and cellulase gene was amplified, purified and sequenced both from native and mutant *A. niger*. On comparison of gene sequences, it was observed that 5 nucleotide base pairs have been replaced in the mutant cellulase. The mutant recombinant enzyme showed 4.5 times higher activity (428.5 µmol/mL/min) as compared to activity of native enzyme (94 µmol/mL/min). The mutant gene was further investigated using Phyre2 and I-Tesser tools which exhibited 71% structural homology with Endoglucanase B of *Thermoascus aurantiacus*. The root mean square deviation (RMSD), root mean square fluctuation (RMSF), solvent accessible surface area (SASA), radius of gyration (Rg) and hydrogen bonds analysis were carried at 35˚C and 50˚C to explore the integrity of structure of recombinant mutant endoglucanase B which corresponded to its optimal temperature. Hydrogen bonds analysis showed more stability of recombinant mutant endoglucanase B as compared to native enzyme. Both native and mutant endoglucanase B genes were expressed in *pET 28a+* and purified with nickel affinity chromatography. Theoretical masses determined through ExPaSy Protparam were found 38.7 and 38.5 kDa for native and mutant enzymes, respectively. The optimal pH and temperature values for the mutant were 5.0 and 50˚C while for native these were found 4.0 and 35˚C, respectively. On reacting with carboxy methyl cellulose (CMC) as substrate, the mutant enzyme exhibited less $K_m$ (0.452 mg/mL) and more $V_{max}$ (50.25 µmol/ml/min) as compared to native having 0.534 mg/mL as $K_m$ and 38.76 µmol/ml/min as $V_{max}$. Among metal ions, $Mg^{2+}$ showed maximum inducing effect (200%) on cellulase activity at 50 mM concentration followed by $Ca^{2+}$ (140%) at 100 mM concentration. Hence, expression of a recombinant mutant cellulase from *A. niger* significantly enhanced its cellulytic potential which could be employed for further industrial applications at pilot scale.

**Data Availability Statement:** All relevant data are within the manuscript and its Supporting Information files.

**Funding:** The research work was funded by an NRPU Project No. 6485, granted by Higher

Education Commission (HEC), Government of Pakistan. The funders had no role in study design, data collection and analysis, decision to publish, or preparation of the manuscript.

## Introduction

Cellulases have garnered significant attention from researchers worldwide due to their wide range of applications across various industries, including starch processing, fermentation of plant products, malting, extraction of vegetable and fruit juices, paper recycling, incorporation as animal food additives and utilization in the fabric industry, rendering it ranked third globally. Keeping in view the wide use of cellulase in industry, there is need for better and efficient cellulase [1,2].

The significance of cellulase production stems from its remarkable role in cellulosic ethanol production. However, it is anticipated that the integration of advanced technology in the future will result in a decrease in the projected selling price of cellulosic ethanol, making it more affordable. [3–5]. Endo glucanase (endo-1, 4—D-glucanase, EG-E.C. 3.2.1.4), beta glucosidase (BG-E.C. 3.2.1.21), and exoglucanase (exo 1, 4—D-glucanase, CBH E.C. 3.2.1.91) make up the cellulase complex that synergetically degrade the celllose [1,6]. Among these enzymes, endo glucanase is considered to be the most efficient [7]. No single microorganism possesses the ability to generate a comprehensive and well-rounded assortment of enzymes that can effectively break down various forms of lignocellulosic biomass [8,9]. *Aspergillus niger* can grow at wide pH (1.4–9.8) and in different range of temperature (6–47°C) making it highly resistant at both acidic and basic pH but its optimal pH is 6 [10].

Impact of genetic alterations on the physiochemical, kinetic and thermodynamic attributes of a highly active glucosidase mutant from *A. niger*, pertaining to the activity and stability of β-glucosidase have been studied. The thermodynamic parameters indicated that the mutant enzyme exhibited greater efficiency and thermostability compared to the parent enzyme [11]. A notable increase in the excretion of various enzymes in *A. niger e.g.* alpha-l-arabinofuranosidase, alpha-glucosidase C, beta-mannosidase and endoglucanase, in addition to glucan 1,4-alpha-glucosidase has been observed at specific pH [12]. Utilization of ultrasound has shown a notable reduction in enzyme activity [13]. Notable variations in glucose yields and ethanol production have been observed based on the specific pretreatment conditions employed [14].

High cost of production is the bottle-neck against the utilization of cellulases in industries. The use of cellulosic material in commercial cellulase manufacturing from fungi has been extensively accepted to lower enzyme production costs [15]. Each year, Pakistan generates over 50 million tons of cellulosic residues that might be used to make cellulases in large quantities [16]. Although efforts have been reported to produce cellulase from microbial sources, however studying the effect of physical, chemical and combined mutagenesis on an indigenous strain of *A. niger* followed by cellulase expression and exploring mutations through bioinformatics parameters leading to multifold enhanced (4.5 times) enzyme activity, demonstrates the significance of our work.

## Materials and methods

### Expression of endoglucanase B in *E. coli* BL21 codon plus

The physical, chemical and combined mutagenesis were incorporated into an indigenous strain of *A. niger* in our previous study to screen the best cellulytic mutant [17]. The selected mutant of *A. niger* was further grown for RNA isolation followed by cDNA synthesis. The primers used for the amplification and cloning of cellulase gene from *A. niger* have been described in S1 Table. The native and mutant cellulase genes were amplified through PCR, the genes were purified and sequenced. Both native and mutant genes were ligated in *pET 28a+* vector for expression studies. The primary culture was cultivated in a 20 mL volume of Luria-

Bertani (LB) growth medium overnight, with the addition of kanamycin (12 μg/20 mL) from a stock solution of 100 μg/mL. The cultivation was carried out at a temperature of 37°C and a rotational speed of 120 rpm. On the next day, a portion equivalent to 2% of the primary over-grown medium was transferred to new flasks with a volume of 1 liter. The flasks were then incubated at a temperature of 37°C and a speed of 120 rpm for a duration of 2–3 hours, until the optical density at a wavelength of 600 nm reached a range of 0.6–0.7. The cultures were treated with a concentration of 50 μM IPTG and subjected to incubation at a temperature of 37°C and a speed of 120 rpm for a duration of overnight. The culture that had been overgrown was exposed to centrifugation at a speed of 6000 rpm for a duration of 10 minutes. The resulting pellets were then treated with sonication for a period of 10 minutes, followed by centrifugation at a speed of 12000 rpm for 20 min at a temperature of 4°C. The fractions were subjected to analysis using SDS-PAGE.

## Purification of recombinant mutant and native endoglucanase B by Ni-column

The recombinant mutant and native form of endoglucanase B were subjected to incubation with a Ni-NTA column that had been prewashed with a solution containing 20 mM Tris-Cl at a pH of 8.0. This incubation process was carried out for a duration of 1 hour at a temperature of 4°C. The protein that was not bound, was subjected to a washing process using a solution containing 2 mM imidazole in 20 mM Tris-Cl at a pH of 8.0. The elution process was conducted using a gradient of imidazole concentrations ranging from 50 to 500 mM, in a solution containing 20 mM Tris-Cl at pH 8.0. The fractions that were obtained were subjected to analysis using a UV spectrophotometer and 10% SDS-PAGE.

## Characterization of enzyme

The native and recombinant cellulase was characterized through determination of following parameters;

## Influence of varying substrate concentration

The optimal concentration of substrate was determined by measuring the activity of cellulase production using different substrates like alpha glucan, cellobiose, CMC at concentrations of 1, 2, 3, 4 and 5%. Reducing sugars were assessed by the DNS reagent method [18].

## Influence of variable pH

The activity of native and recombinant enzyme was measured at different pH values e.g., 4, 5, 6, 7 and 8 with NaOH and HCl. DNS assay was followed to estimate the reducing sugars [18].

## Influence of variable temperature

The activity of native and recombinant enzyme was determined at variable temperature like 20, 25, 30, 35, 40, 45, 50, 55, 60, 65 and 70°C. DNS assay was followed for the determination of activity of cellulase [18].

## Impact of metal ions on cellulase

The impact of metal ions on enzyme was measured by typical assay method in the existence of ions such as $Zn^{2+}$, $Cu^{2+}$, $Hg^{2+}$, $Ni^{2+}$, $Mg^{2+}$, and $Ca^{2+}$. These ions were tested as metal chloride at 5mM [19].

## Determination of $K_m$ and $V_{max}$ values

The $K_m$ and $V_{max}$ values of the cellulase enzyme were obtained using the Lineweaver-Burk plot method. This included analyzing the reaction rates at various concentrations of cellulase, ranging from 0 to 20 mg/mL. The cellulase enzymes of native and mutant *A. niger* were compared by a process of characterization. The study aimed to identify the impact of pH, optimal temperature, optimal substrate concentration, activators, and inhibitory metal ions on the observed phenomena. The Michaelis-Menten constant ($K_m$) and the maximum velocity ($V_{max}$) were found.

## Molecular dynamic simulations (MDS)

To duplicate the cellular environment, a dodecahedron-shaped aqueous container was used, consisting of TIP3P water molecules and sodium and chloride ions, to enclose the complex system. To guarantee the system's stability, a steepest descent algorithm including energy efficiency concerns was used over a time span of 200 picoseconds. Subsequently, thermodynamic equilibrium was attained by the use of the CHARMM36 force field and GROMACS software. A molecular dynamics simulation with a duration of 100 nanoseconds was then conducted, whereby a long-range Van der Waals cut-off rvdm, created during the energy minimization phase was included [20].

The simulation was started by setting the starting temperature to 300 Kelvin and using a generation seed of -1, while applying the generation velocity option. The Parrinello-Rahman approach, a pressure coupling technique, was used at the last stage of the molecular production dynamics. The parameter tau p was assigned a value of 2.0, which corresponds to the relaxation time for pressure. Similarly, the parameter ref p was assigned a value of 1.0, indicating the reference pressure for the system. The assessment of hydrogen bonding calculations and simulation results included the use of several computational tools, such as PyMOL and VMD. The analysis was performed using many built-in modules of GROMACS, such as gmx rms, gmx rmsf, gmx area, and gyrate [21].

The study of binding interactions was carried out using mmPBSA analysis, a computational technique often applied in molecular biology investigations. The technique yielded a heat map that visually depicted the binding affinity of specific residues. The following is a comprehensive guide on the analysis of protein-protein complexes using several metrics.

## RMSD (Root-mean-square-deviation)

The root-mean-square deviation (RMSD) is a commonly used measure within the discipline of structural biology for evaluating the level of similarity and stability shown by protein-ligand complexes. The average distance between the corresponding atoms of the superimposed structures was quantified, yielding significant insights into the structural changes that occur as a result of the ligand binding to the protein. Through the utilization of the root mean square deviation (RMSD) metric, scholars are able to assess the efficacy of the complex and ascertain the degree to which the ligand interacts with the protein. This facilitates the quantification of RMSD by means of comparing two structures, which can be derived from molecular dynamics simulations or refined through techniques in structural chemistry. The presence of a decreased root-mean-square deviation (RMSD) between a protein and its corresponding ligand has been considered as a positive sign of their shared structural similarity and stability [22,23].

## RMSF (Root mean square fluctuation)

The root mean square fluctuation (RMSF) is a significant statistic used in the field of protein-ligand system simulation and optimization. The metric described above measured the root

mean square deviation of the spatial coordinates of an atom relative to its average location within a protein-ligand combination. The use of root-mean-square fluctuation (RMSF) analysis allowed for the discernment of certain areas within a protein that experience varying degrees of impact upon the binding of a ligand.

The present study offered significant information into the degree of protein flexibility. Areas exhibiting elevated root mean square fluctuation (RMSF) values were indicative of heightened flexibility, while areas displaying lower RMSF values were suggestive of increased stiffness [22,23].

## SASA (Solvent-accessible surface area)

The metric known as solvent accessible surface area (SASA) was used to measure the degree of interaction between individual solvent molecules and a complex formed by a protein and ligand. This research paper provided an examination of the structural soundness of the system and investigated the complicated interplay between the complex and its surrounding environment. The solvent-accessible surface area (SASA) of a molecule served as a metric for evaluating its degree of compactness or elongation. A low solvent-accessible surface area (SASA) value indicated a molecule that was densely arranged, with few exposed surfaces. On the other hand, a high SASA (solvent-accessible surface area) value indicated a molecule that exhibits elongation or stretching, resulting in an increased number of exposed surfaces [24].

## Radius of gyration (Rg)

The radius of gyration (Rg) was a quantitative parameter used to assess the distribution of atomic distances inside a protein-ligand complex relative to its geometric center of mass. The quantification of compactness within a complex served as a significant means of acquiring knowledge on the influence of a ligand on the overall structure of a protein. The determination of the radius of gyration (Rg) provided valuable insights on the conformational properties of a molecule. According to Ahamad et al. (2022), a higher Rg value indicates a molecular configuration that is extended or elongated, whereas a lower Rg value suggests a molecular structure that is more compact or condensed [24].

## Hydrogen bonds

Hydrogen bonds were of significant importance in the quantification of protein-ligand interactions, as they aided in the understanding of the specificity and strength of the intermolecular relationships between these two entities.

## Results and discussion

### Cloning and expression of mutant and native cellulase

The mRNA was extracted from native and mutant strains yielding maximum cellulase activity. The extracted mRNA of both native and mutant strains of *A. niger* has been shown in S1 Fig. The cDNA was synthesised and both native and mutant cellulase genes were amplified using PCR (S2 and S3 Figs). The amplified gene was sequenced and protein sequence alignment of native and mutant endoglucanase was performed and shown in S4 and S5 Figs, respectively. The gene was cloned and expressed in *pET*28a(+) vector. The expression was confirmed through double digestion of vector as shown in S6 Fig. The soluble expression of mutant and native cellulase was analyzed by SDS-PAGE after 10 min of sonication. From SDS-PAGE soluble expression was obtained in 1st supernatant fraction after sonication followed by centrifugation. Different volumes of native and mutant endoglucanase B supernatants were analyzed by

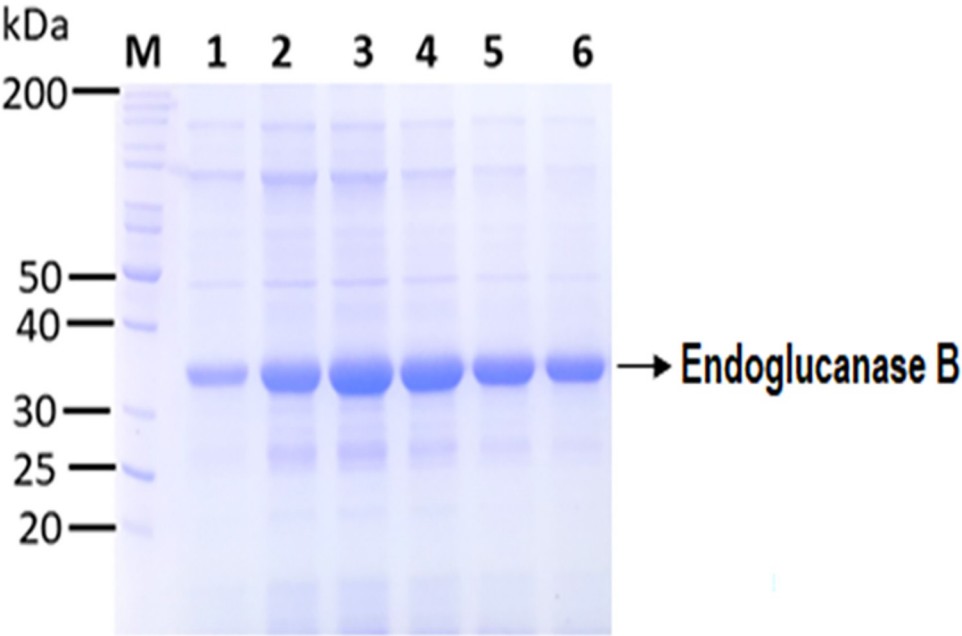

**Fig 1. SDS-PAGE analysis of soluble native and mutant endoglucanase B expression from the recombinant *E.coli* BL21 (DE3).** Lane M: Mobilities of proteins of known molecular masses on SDS-PAGE. Lane 1–3: 5, 10 and 20 μL of native endoglucanase B supernatant obtained after sonication respectively. Lane 4–6: 20, 10 and 5 μL of mutant endoglucanase B obtained after sonication respectively.

SDS-PAGE. Mutant endoglucanase B had less bacterial cellular protein content as compared to native one (Fig 1).

Both native and mutant endoglucanase B were observed at similar size position. Theoretical masses were calculated from ExPaSy Protparam tool and it was 38.7 and 38.5 kDa for native and mutant enzyme respectively although mutant endoglucanase B had multiple mutations as described below (S5 Fig) however all the mutations were substitution and few were deletion at C terminal of mutant endoglucanase, therefore the whole mass of protein was close to native one.

### Purification of native and mutant endoglucanase B by Ni-affinity chromatography

The native and mutant endoglucanase B were purified with Ni-affinity chromatography. Firstly about 250 mL of mutant endoglucanase B containing 2521 mg of total protein content were loaded to Ni column. Collected flow through were UV quantified and it was analyzed that most of protein bounded to column. Mutant endoglucanase B was eluted from 200–250 mM imidazole. The maximum amount of protein eluted at 250 mM imidazole (S7 Fig). Similarly, native endoglucanase B was proceeded. Native endoglucanase B followed almost similar pattern of elution as did mutant endoglucanase B. The purified native enzyme was eluted at 250 mM imidazole (S8 Fig).

The purified native mutant endoglucanase B was observed around 39 kDa protein size which was marked by protein marker. The native endoglucanase B has theoretical size of 36.5 kDa (O74706 · EGLB_ASPNG) from https://www.uniprot.org/uniprotkb/O74706/. The theoretical masses of enzyme sequence containing His-Tag at N-Terminal were calculated ExPaSy Protparam and theoretical mass was 38.5 kDa for both native and mutant endoglucanase B. The purified fractions of native and mutant endoglucanase B were combined and dialyzed against 20 mM Tris-HCl, dialyzed enzymes were analyzed by SDS-PAGE (Fig 2).

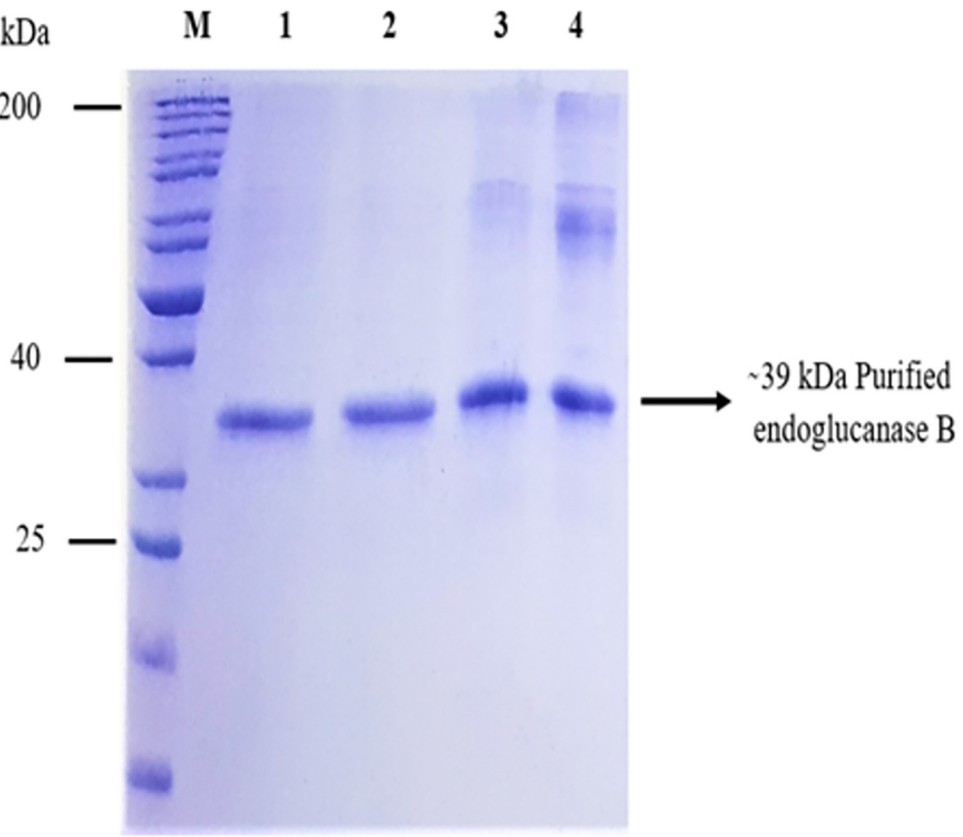

**Fig 2. Purified mutant and native endoglucanase B analyzed by SDS-PAGE.** M: Mobilities of proteins of known molecular masses on SDS-PAGE. Lane 1–2: Showing purified dialysed mutant endoglucanase B. Lane 3–4: Showing purified dialysed native endoglucanase B.

## Characterization of mutant endoglucanase B

Initially the potential mutant endoglucanase B was ruled out among other mutants based upon its activity. The activity was determined by glucose standard curve which was constructed by using DNS method. The observed activity of mutant endoglucanase B was 428.6 μmol/mL/min which was quite effective as compared to native endoglucanase B 94 μmol/mL/min.

## Effect of pH on endoglucanase B activity

Enzyme activity was observed at a wide pH range (2–9) for both native and recombinant mutant endoglucanase B. In a holistic view, enzyme activity for mutant endoglucanase B was higher than native one at observed pH range. Enzyme activity of native endoglucanase B gradually increased till pH 4.0 and then decreased subsequently by increasing pH, and it was very low at pH 9.0. Whereas the enzyme activity of mutant endoglucanase B was gradually increased by increasing the pH till pH 5.0 and then decreased but still had significant enzyme activity as compared to native one. The optimum pH for native and mutant endoglucanase B was 4 and 5 respectively (Fig 3).

## Effect of temperature on endoglucanase B activity

Temperature effect on enzyme activity was observed at 20–70˚C for both native and mutant endoglucanase B (Fig 4). Maximum enzyme activity was observed at 35˚C for native and it was

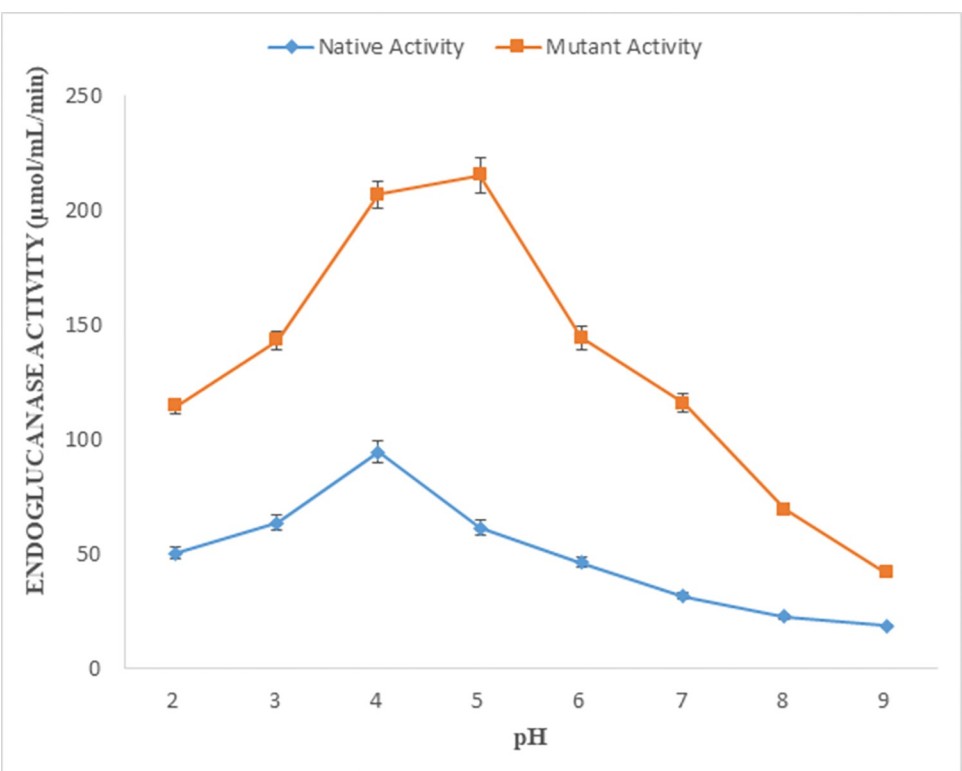

**Fig 3. Effect of pH on enzyme activity of native and mutant endoglucanase B from *A. niger*.**

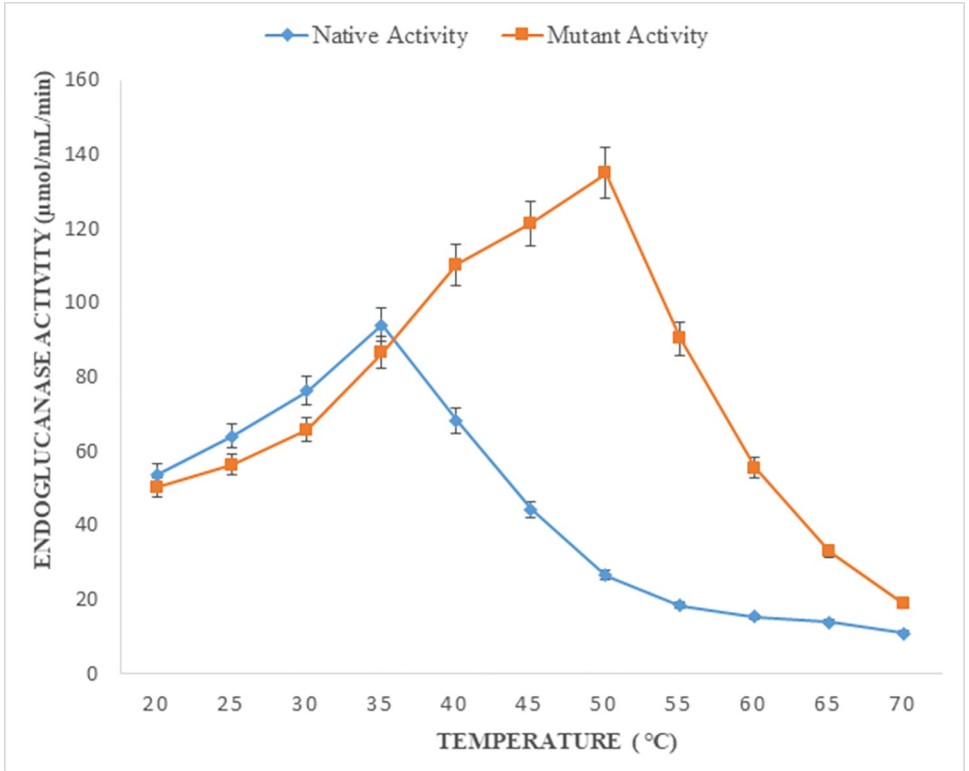

**Fig 4. Effect of temperature on enzyme activity, native and mutant endoglucanase B showed optimal activity at 35˚C and 50˚C respectively.**

gradually decreased by increasing the temperature. Whereas recombinant mutant enzyme activity gradually increased till 50°C and decreased gradually by further increase in temperature.

Till 35°C, native enzyme had slightly greater activity however by the increase of temperature, recombinant mutant endoglucanase B activity significantly increased. At 50°C, enzyme activity of mutant endoglucanase B was 5 folds higher than its native form.

## Substrate specificity of purified native and recombinant mutant endoglucanase B

Following substrates were studied to evaluate substrate specificity Cellobiose, Alpha glucan and carboxymethyl cellulose (CMC). For alpha glucan and cellobiose, both native and mutant endoglucanase the enzyme activities were 53 and 68 μmol/mL/min, respectively and there was no significant statistical difference found in the enzyme activity for either of the substrates. Whereas mutant endoglucanase B exhibited more activity (428.5 μmol/mL/min) for CMC than native enzyme. While native endoglucanase B had 94 μmol/mL/min activity for CMC although the activity was increased than other two substrates but still it was quite lower than mutant endoglucanase B (Fig 5).

Both native and mutant endoglucanase B had substrate specificity for carboxymethylcellulose (CMC) and mutant endoglucanase B had 428.5 μmol/mL/min enzyme activity.

## The impact of substrate concentration on the activity of endoglucanase B

The impact of substrate concentration, namely CMC, was evaluated using the principles of Michaelis-Menten Kinetics. To establish the kinetic parameters Km and Vmax, a Lineweaver-Burk Plot was built. The kinetic parameters, specifically the Michaelis-Menten constant ($K_m$) and the maximum reaction rate ($V_{max}$), were determined for both native and recombinant

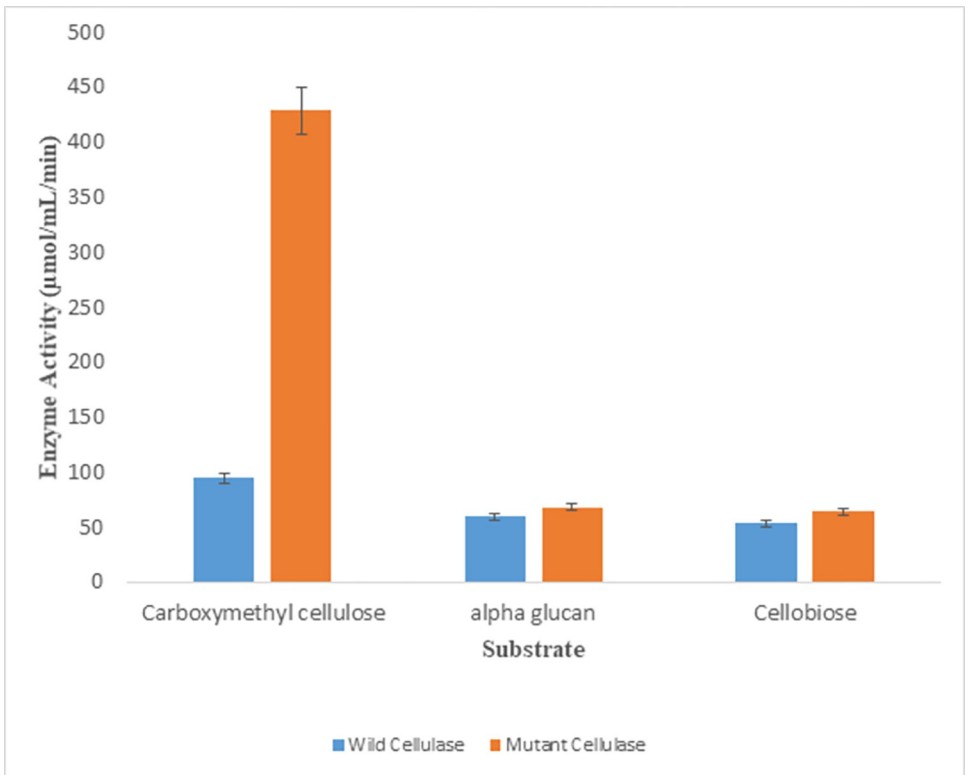

**Fig 5. Substrate specificity for native and recombinant mutant endoglucanase B.**

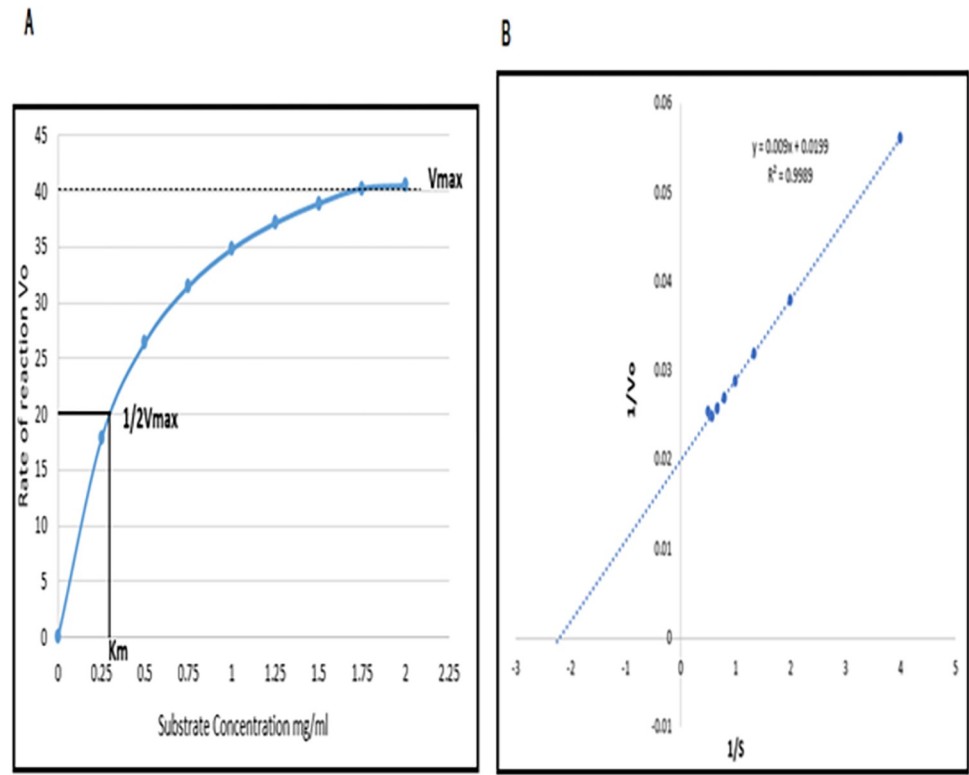

**Fig 6. Enzyme kinetics studies for mutant endoglucanase B by Michaelis-Menten Kinetics and Lineweaver-Burk.**

mutant endoglucanase B. The $K_m$ value was found to be 0.452 mg/mL, while the $V_{max}$ value was determined to be 50.25 μmol/mL/min for recombinant mutant endoglucanase B. These values were obtained from the Lineweaver-Burk plot (Fig 6). The initial investigation involved the application of Michaelis-Menten Kinetics, wherein the observation of $V_{max}$ occurred at a concentration of 2 mg/mL of CMC. Additionally, the concentration of substrate at which half of the maximum velocity (1/2 $V_{max}$) was observed was determined to be 0.275 mg/mL, commonly referred to as $K_m$.

The $K_m$ and $V_{max}$ values for native endoglucanase B were determined to be 0.3 mg/mL of substrate and 31 rate of reaction, respectively. The values for $K_m$ and $V_{max}$ were determined to be 0.534 mg/mL and 38.76 μmol/mL/min, respectively (Fig 7). The $K_m$ value of the mutant endoglucanase was lower, while the $V_{max}$ value was higher when compared to the native endoglucanase B.

## Effect of various metal ions on endoglucanase B activity

Following divalent metal ions Mg, Ca, Zn, Cu, Hg and Ni having concentrations 50–250 mM, were used with common ion chloride (Cl) to study for their effectiveness on enzyme activity and percentage relative activity was calculated by comparing to control which did not contain any metal ion. Activity at control was taken as 100%, for mutant endoglucanase B, % relative activity significantly increased from 143–200% and was maximum at 50 mM of Mg ions. In case of Ca ions, the % relative activity was increased 115–140% and it was maximum at 100 mM of Ca ions. Whereas % activity gradually decreased in case of Cu and Hg ions up to 86% and it was also decreased up to 65% for Zn ions by increasing the metal ion concentration. In case of Ni ions, % activity significantly decreased up to 50% by increasing ionic concentration (Fig 8).

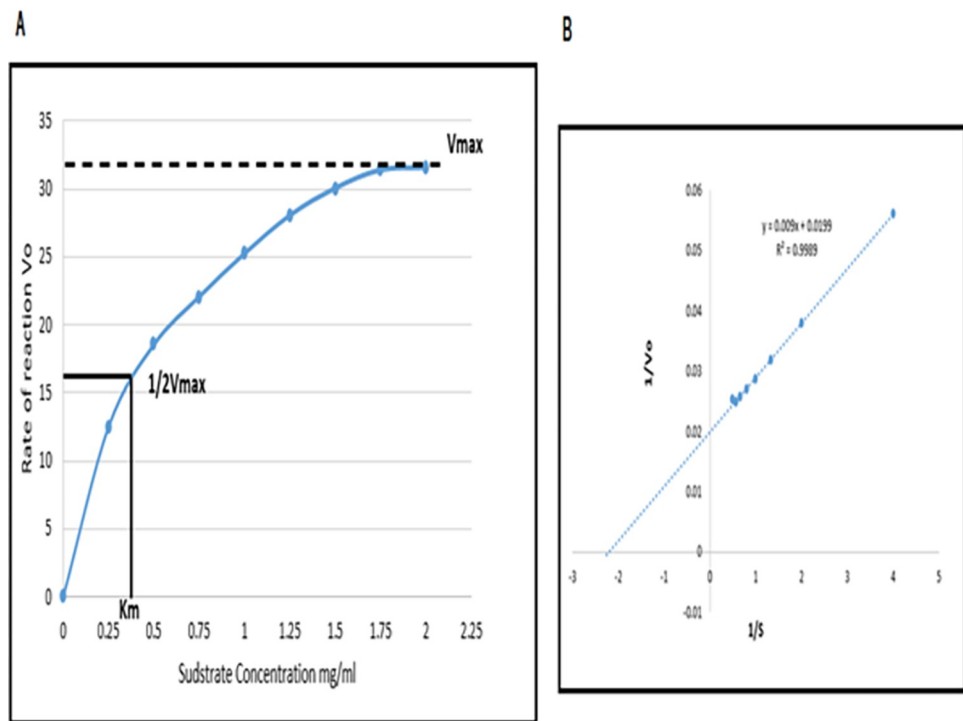

**Fig 7. Enzyme kinetics studies for native endoglucanase B by Michaelis-Menten Kinetics and Lineweaver-Burk.**

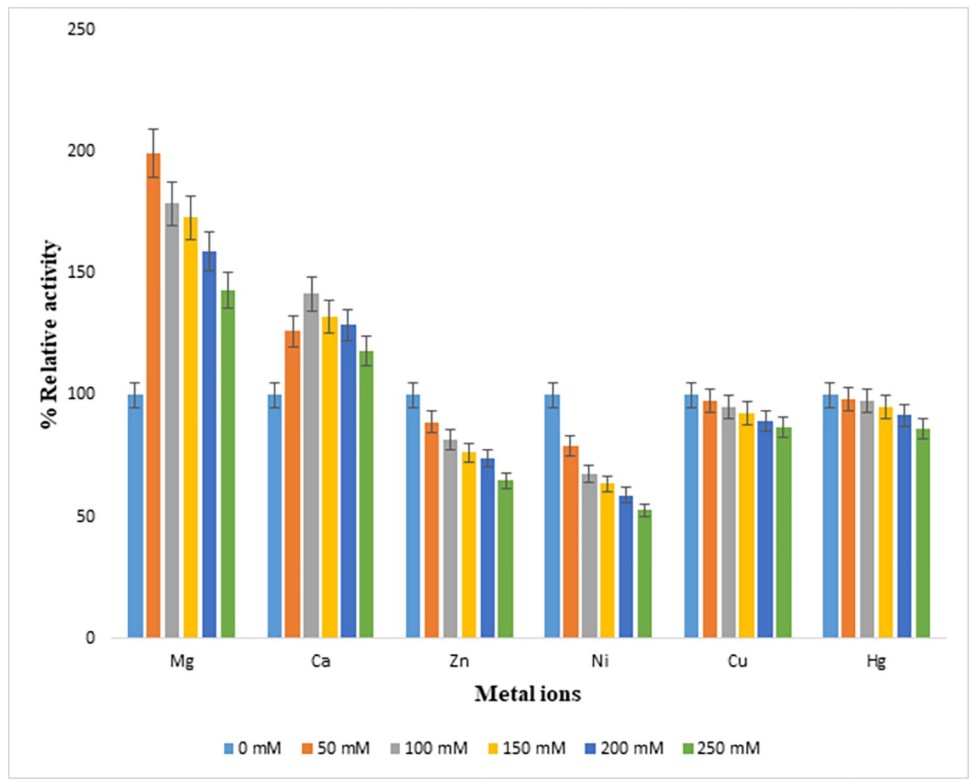

**Fig 8. Effect of metal ion concentrations on % relative activity of recombinant mutant endoglucanase B from *A. niger*.**

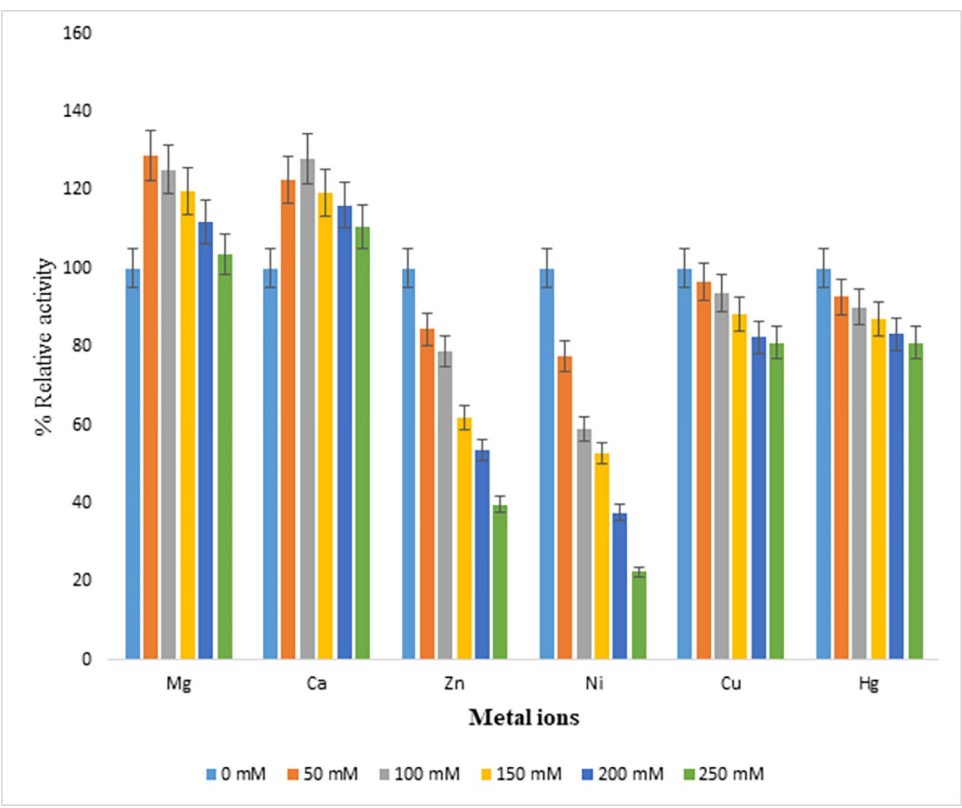

**Fig 9. Effect of metal ion concentrations on % relative activity of native endoglucanase B from *A. niger*.**

Similarly metal ions were also studied for native endoglucanase B at 50–250 mM concentrations, $Mg^+$ and $Ca^+$ ions posed increased % relative activity 103–129% and maximum activity was observed at 50 and 100 mM of $Mg^+$ and $Ca^+$ ions, respectively. There was decrease in % relative activity up to 96–81% by increasing ion concentration in case of $Hg^+$ and $Cu^+$ ions. While in case of Zn and Ni ions, % relative activity was significantly decreased up to 84–22% by increasing ionic concentrations (Fig 9).

By comparing the metal ion effect on % relative activity of native and recombinant mutant endoglucanase B, the activities of mutant endoglucanase B were higher than native one although the increasing and decreasing pattern of relative activities was similar for all metal ions.

### Bioinformatics studies of mutant endoglucanase B

The secondary structure of mutant endoglucanase B was predicted by Phyre2, there was 29% alpha helix, 13% beta sheets and 4% disordered structure. Whereas 3D structure was predicted by Phyre2 and I-Tesser. The 3D structure was identical by both of software and it had 71% structural similarity with endoglucanase B of *Thermoascus aurantiacus* with C-score -1.53, TM score 0.861 and RMSD value 0.35. The structure variation was analyzed by super imposing template and target structure (Fig 10).

The predicted structure of mutant endoglucanase B was rich in alpha helix and beta sheets was present as barrel form in center. The active site residues are present in beta sheet structure which is predicted by enzyme ligand interaction by I-Tesser in which beta cellutriose is used as ligand (Fig 11). The predicted ligand binding residues are 60, 140, 141, 179, 180, 246, 286

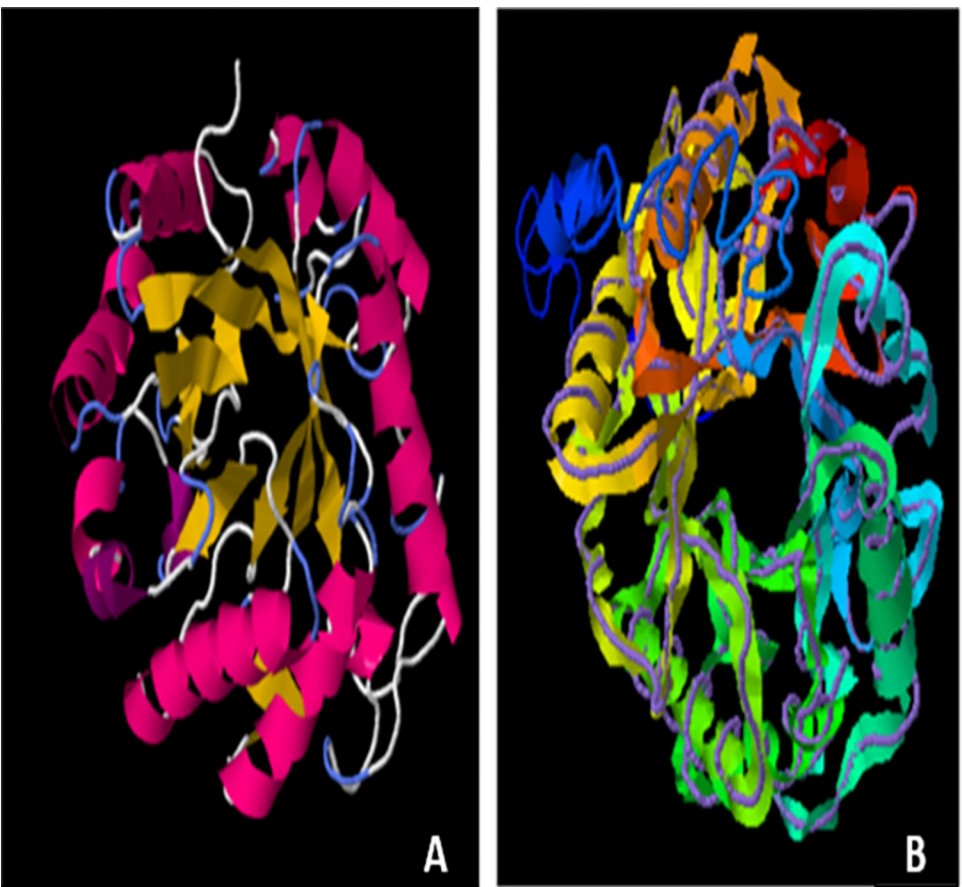

**Fig 10.** 3D predicted structure of endoglucanase B from *A. niger* (A): Predicted structure of endoglucanase B and (B): Superimposed image of endoglucanase B.

and319 while the active site residues are 179, 180, 244 and 286. Although the binding site and active site residues are conserved but the mutated endoglucanase B showed higher activity which can be major attribute of *A. niger* as organism.

## Molecular dynamic simulation

In the final step of molecular production dynamics, the Parrinello-Rahman method was utilized for pressure coupling. The value of tau p was set to 2.0, while the value of ref p was set to 1.0. Hydrogen bonding calculations and simulation results evaluation were performed using a range of tools, including PyMOL and VMD. The study was conducted using GROMACS built-in modules, including gmx rms, gmx rmsf, gmx area, and gyrate.

### Root mean square deviation (RMSD) of wild and mutant type of endoglucanase B

RMSD analysis of mutant and wild type cellulase from *A. niger* at 35˚C and 50˚C (Figs 12 and 13). Wild type of endoglucanase B was stable at 35˚C, while the mutant type of endoglucanase B was stable at 50˚C as represented by RMSD analysis. The RMSD is a metric used to assess the degree of structural similarity and stability shown by a protein-ligand combination. A reduction in the root mean square deviation (RMSD) between a protein and its ligand served as a favorable indication of their structural resemblance and stability.

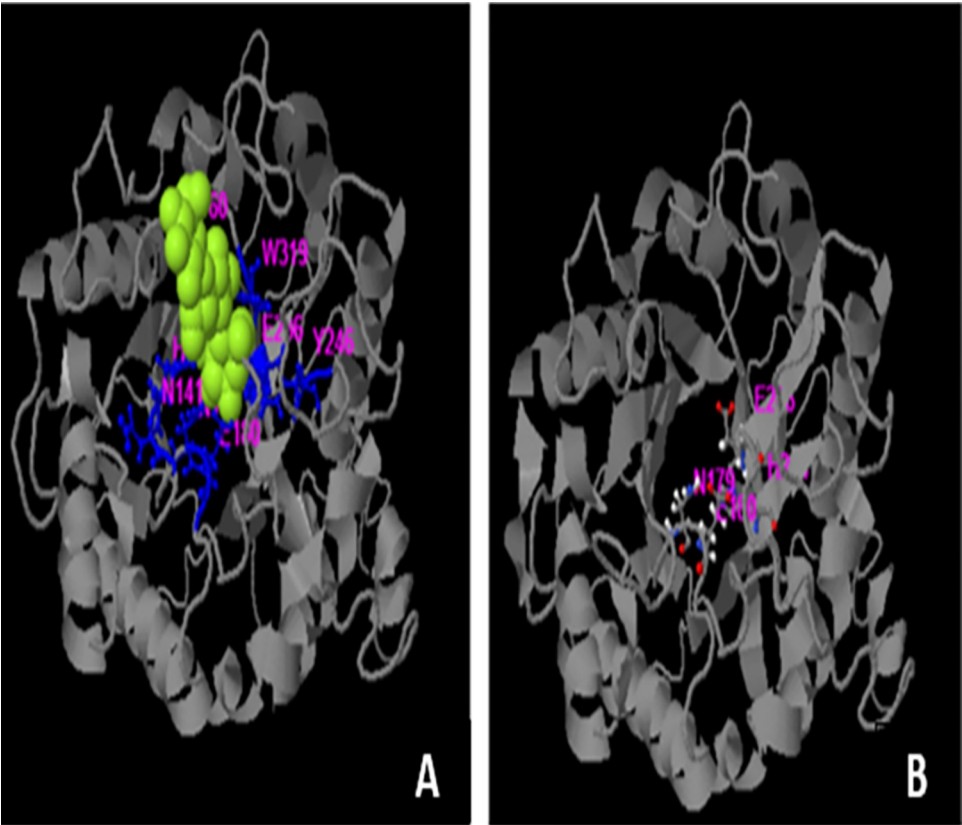

**Fig 11.** Enzyme ligand interaction of endoglucanase B with beta cellurtriose (A) and active site residues (B). Yellow highlighted is ligand molecule and amino acids are highlighted which are involved in enzyme ligand interaction.

## Root mean square fluctuation (RMSF) of wild and mutant type of endoglucanase B

The Root Mean Square Fluctuation (RMSF) is a widely utilized measure in Molecular Dynamics (MD) simulations. This metric quantifies each atom's mean atomic displacement relative to its simulated average position. Scientists use the RMSF metric to calculate the temporal behaviour of a protein or molecule during molecular dynamics (MD) simulations to determine its flexibility or stability. Higher root mean square fluctuation (RMSF) values indicate more atomic mobility, whereas lower values indicate more rigid regions.

RMSF analysis of mutant and wild type cellulase from *A. niger* at 35˚C and 50˚C (Figs 14 and 15). Wild type of endoglucanase B was stable at 35˚C, while the mutant type of endoglucanase B was stable at 50˚C as represented by RMSF analysis.

The metric quantified the root mean square deviation of an atom's spatial coordinated with respect to its average position within a protein-ligand complex. A high root mean square fluctuation (RMSF) value was indicative of a region with more flexibility, whereas a low RMSF value suggested a region with greater rigidity.

## Radius of gyration (Rg) of wild and mutant type of endoglucanase B

The radius of gyration analysis of mutant and wild type cellulase from *A. niger* at 35˚C and 50˚C (Figs 16 and 17). Wild type of endoglucanase B was stable at 35˚C, while the mutant type of endoglucanase B was stable at 50˚C as represented by radius of gyration analysis. The

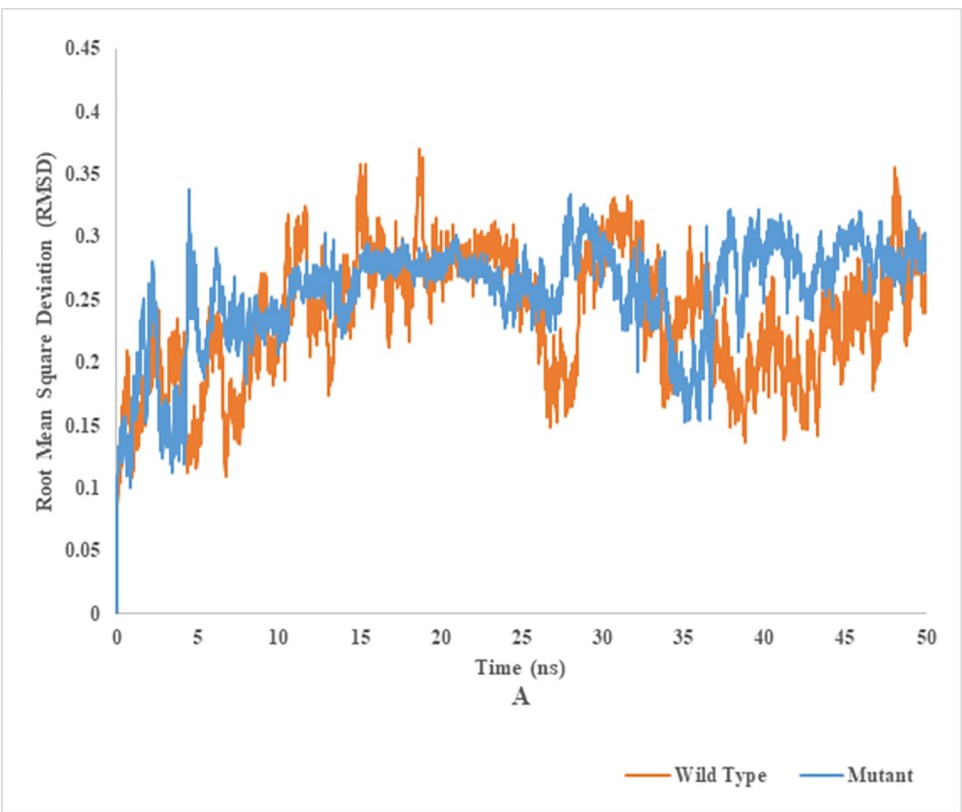

**Fig 12. Root mean square deviation analysis of mutant and wild type of cellulase (endoglucanase B) from** *Aspergillus niger* **at 35˚C.**

measurement of compactness of the complex offered valuable insights into the impact of the ligand on the overall conformation of the protein. A higher value of Rg signified an elongated molecular structure, whereas a lower value of Rg indicated a denser molecular arrangement.

## Solvent accessible surface area (SASA) of wild and mutant type of endoglucanase B

The SASA analysis of mutant and wild type cellulase from *A. niger* at 35˚C and 50˚C have been shown in the Figs 18 and 19, respectively. Wild type of endoglucanase B was stable at 35˚C, while the mutant type of endoglucanase B was stable at 50˚C as represented by SASA analysis.

## Hydrogen bond analysis of wild and mutant type of endoglucanase B

The Hydrogen bond analysis of mutant and wild type cellulase from *A. niger* at 35˚C and 50˚C have been shown in the Figs 20 and 21, respectively. Wild type of endoglucanase B was stable at 35˚C, while the mutant type of endoglucanase B was stable at 50˚C as represented by hydrogen bond analysis.

The predicted ligand binding residues are 60,140,141,179,180,246,286,319 and active site residues are 179,180,244 and 286. Phenylalanine was replaced by proline due to mutation at position 32. The backbone was same for each amino acid but side chain was unique for both amino acids. Each amino acid had its own specific size, charge and hydrophobicity-value so native and mutant residues differ in these properties. The mutant residue was smaller than native residue. The effect of mutation would be on the features like contacts made by the

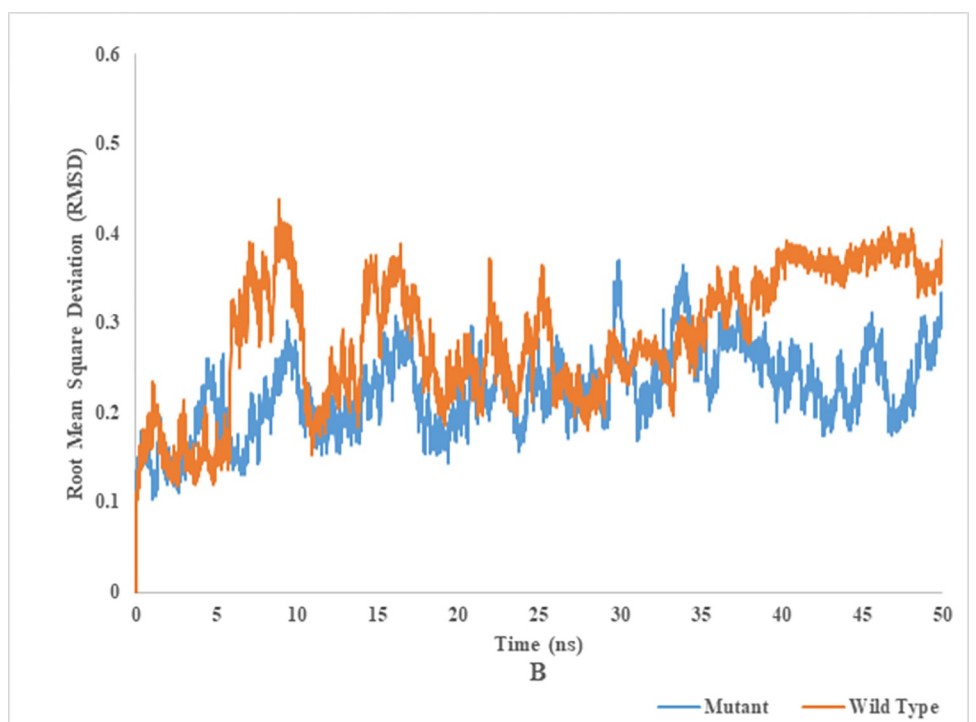

**Fig 13. Root mean square deviation analysis of mutant and wild type of cellulase (endoglucanase B) from *Aspergillus niger* at 50˚C.**

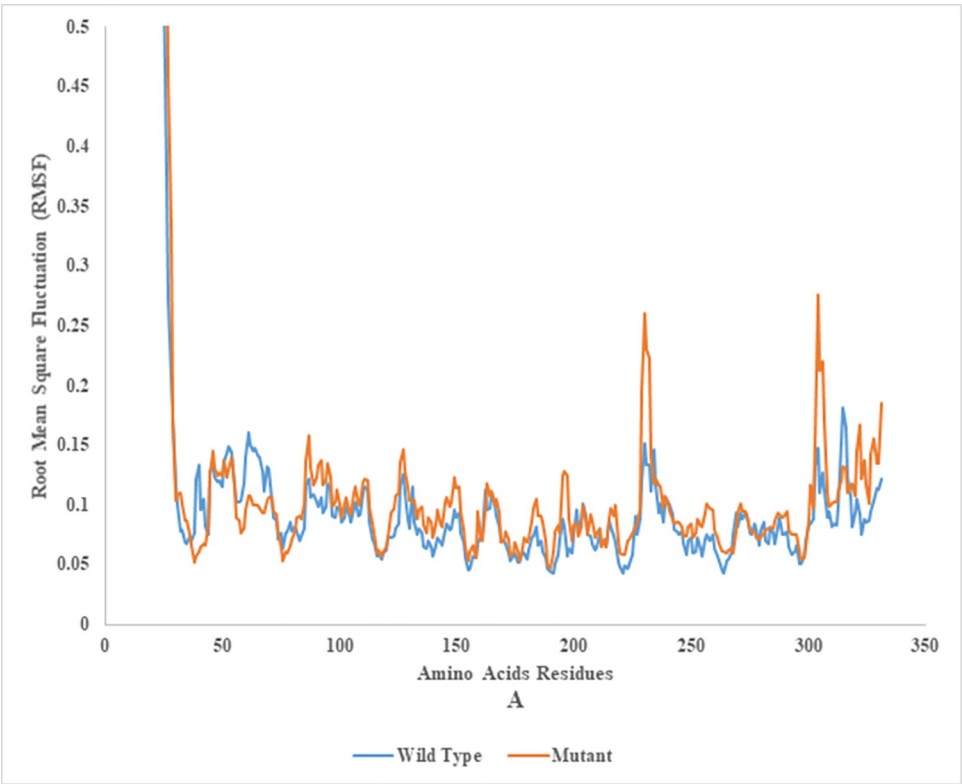

**Fig 14. Root mean square fluctuation analysis of mutant and wild type of cellulase (endoglucanase B) from *Aspergillus niger* at 35˚C.**

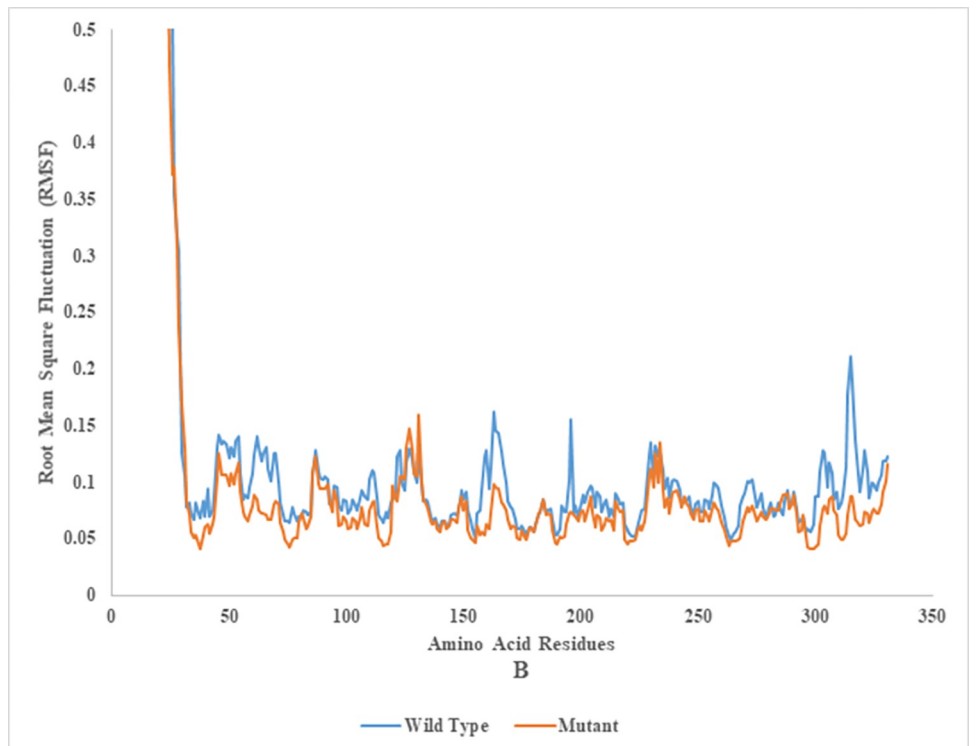

**Fig 15. Root mean square fluctuation analysis of mutant and wild type of cellulase (endoglucanase B) from** *Aspergillus niger* **at 50˚C.**

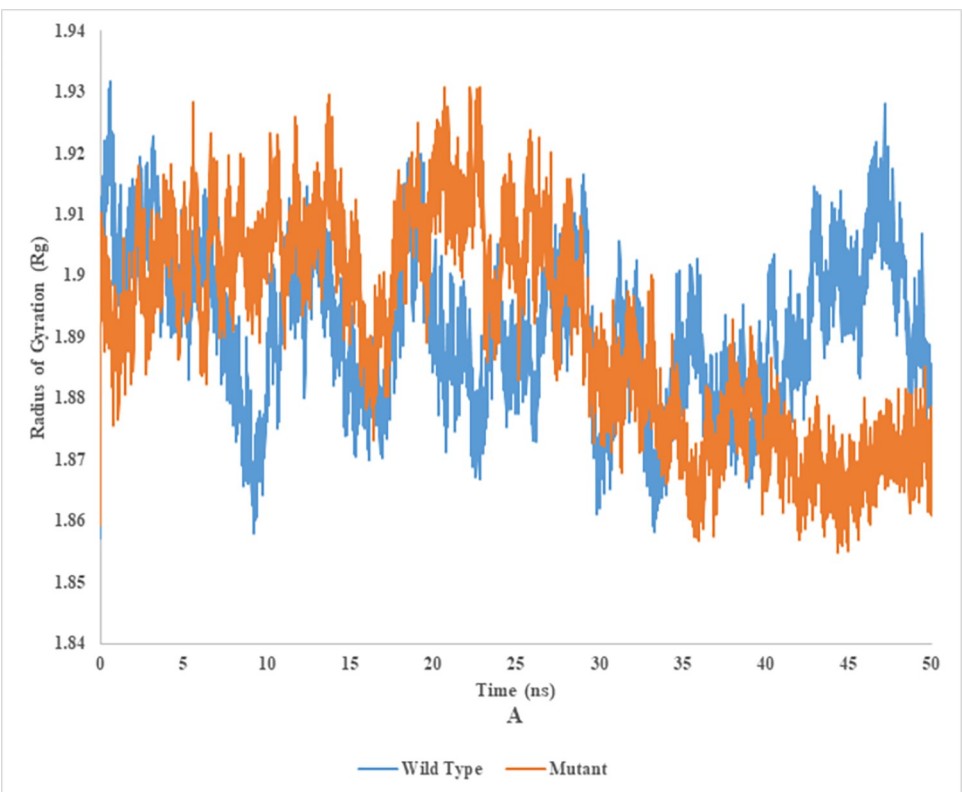

**Fig 16. Radius of gyration analysis of mutant and wild type of cellulase (endoglucanase B) from** *Aspergillus niger* **at 35˚C.**

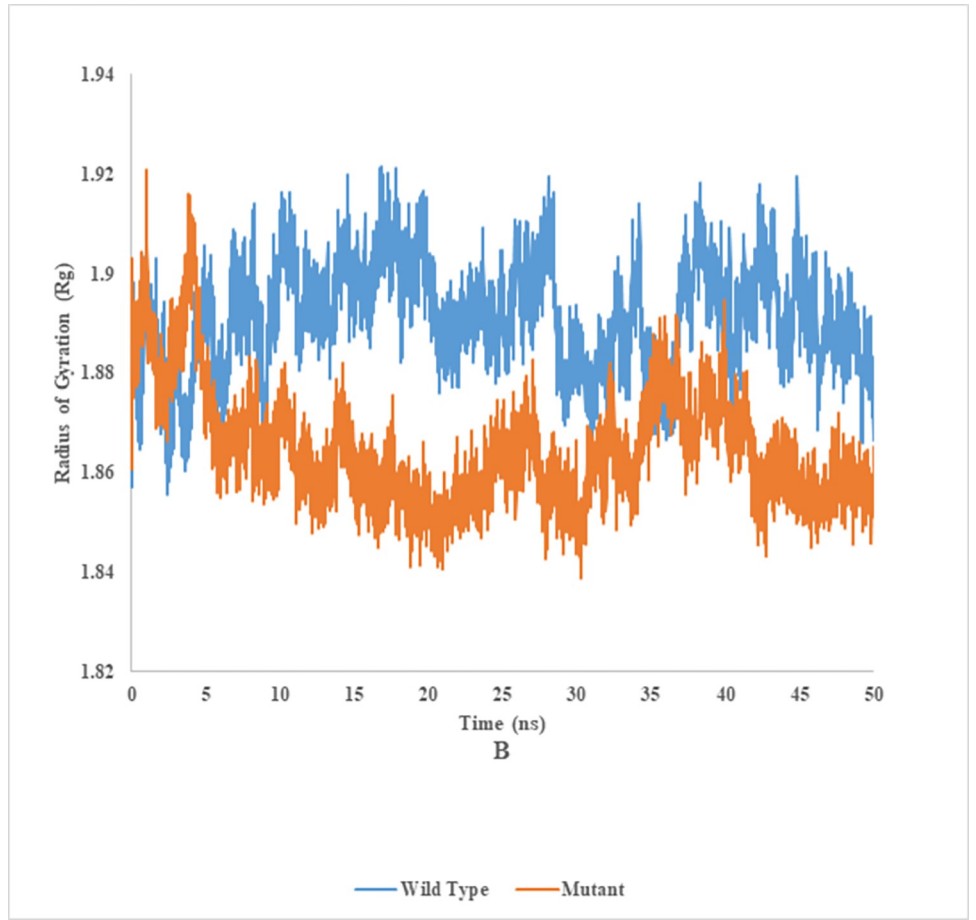

**Fig 17. Radius of gyration analysis of mutant and wild type of Cellulase (endoglucanase B) from *Aspergillus niger* at 50˚C.**

mutated residue, structural domains in which the residue was located, modifications on this residue and known variants for this residue. The wild-type residue was predicted (using the Reprof software, version 1.0.1) to be located in its preferred secondary structure, a β-strand. The mutant residue preferred to be in another secondary structure; therefore, the local conformation would be slightly destabilized. The mutant and wild-type residue were not very similar. The mutation of a valine into a methionine was observed at position 51. The backbone was same for each amino acid but side chain was unique for both amino acids. The mutant residue was bigger than wild-type residue and mutant residue was located near a highly conserved position. The mutated residue was located in a domain that is important for the main activity of the protein. Mutation of the residue might disturb this function. The mutation of an arginine into serine at position 124 in which backbone was the same for each amino acid but the side chain was unique for each amino acid. The mutant residue was smaller than wild-type residue. There was a difference in the charge between wild-type and mutant amino acid. The charge of wild-type residue was lost which might have affected interaction with other molecules or residues. The mutated residue was located in a domain that was important for the main activity of the protein. Mutation of the residue was supposed to disturb this function.

For the function of enzyme following regions are important: active site, binding site, ligand binding site and topological structure of enzyme which contributes in stability and functional

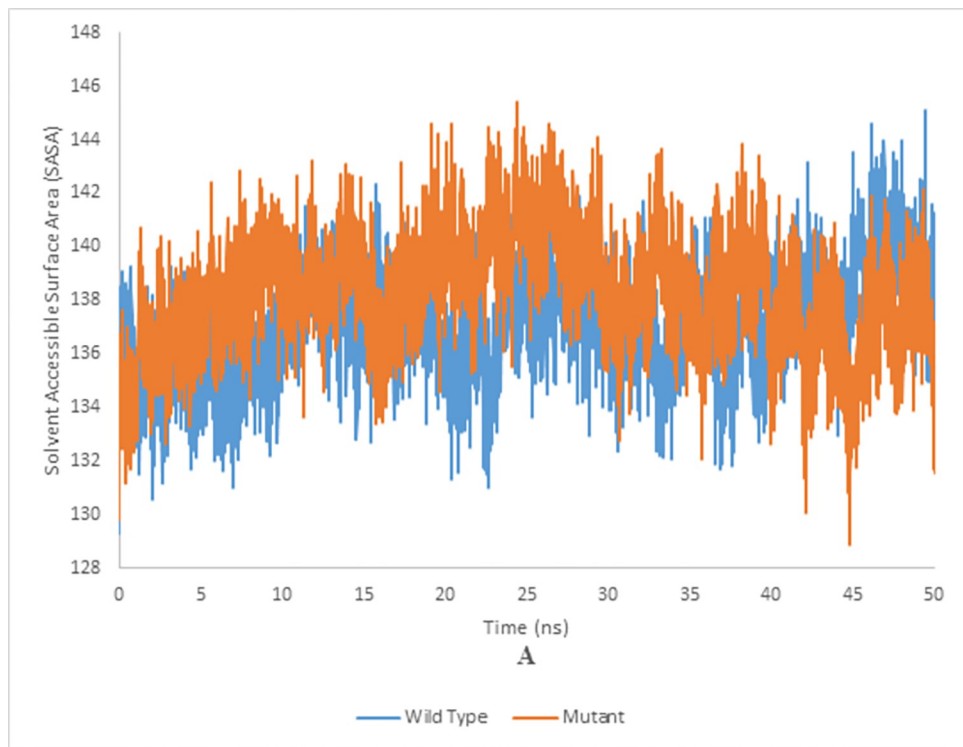

**Fig 18. Solvent accessible surface area analysis of mutant and wild type of cellulase (endoglucanase B) from *Aspergillus niger* at 35˚C.**

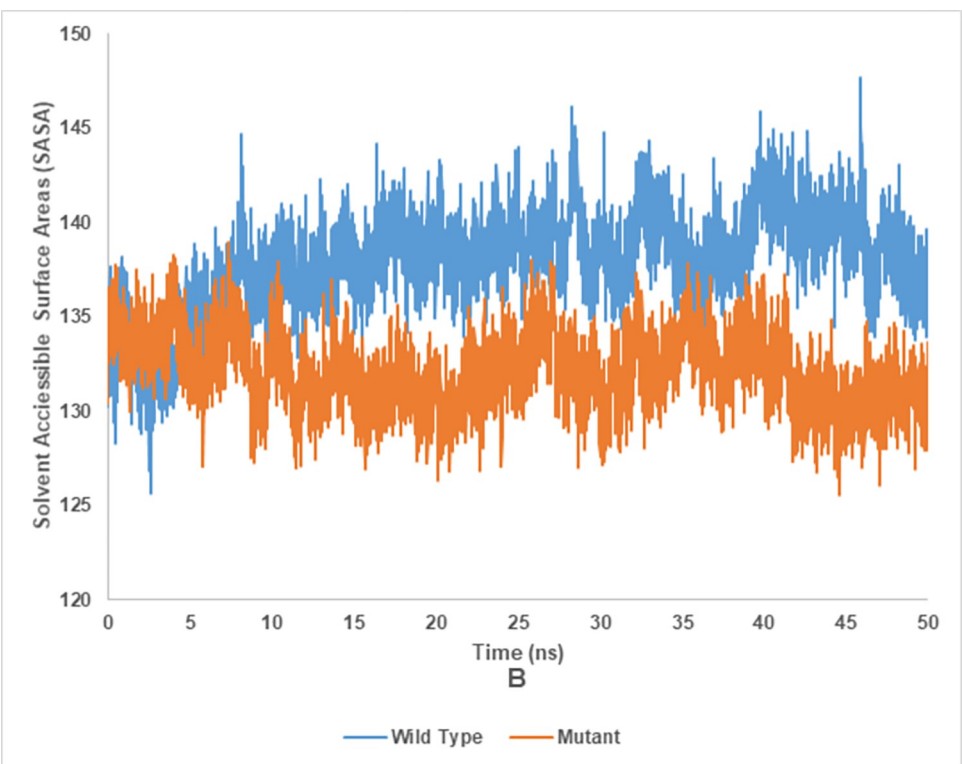

**Fig 19. Solvent accessible surface area analysis of mutant and wild type of cellulase (endoglucanase B) from *Aspergillus niger* at 50˚C.**

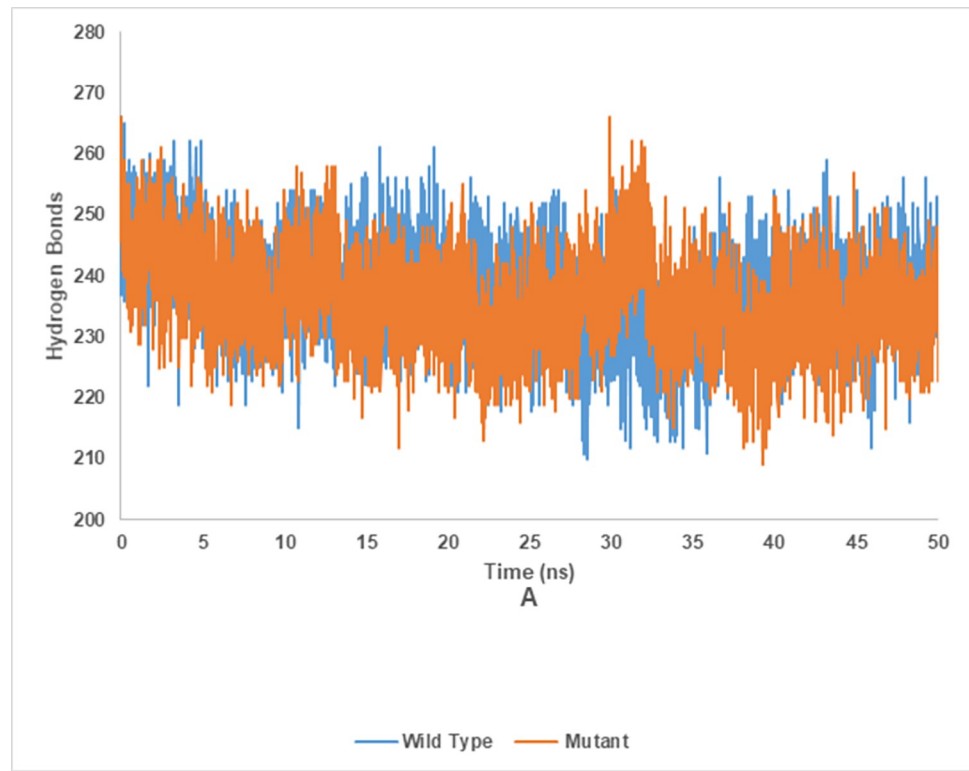

**Fig 20. Hydrogen bond analysis of mutant and wild type of cellulase (endoglucanase B) from *Aspergillus niger* at 35˚C.**

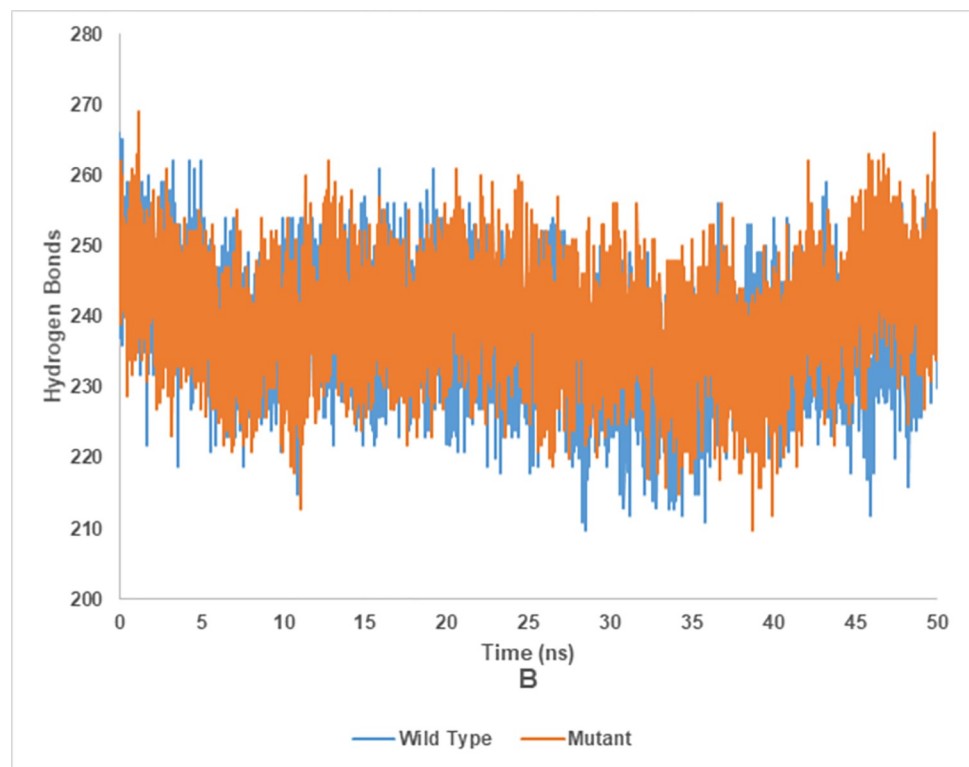

**Fig 21. Hydrogen bond analysis of mutant and wild type of cellulase (endoglucanase B) from *Aspergillus niger* at 50˚C.**

dynamics of enzyme. The site directed mutation was introduced in protease binding site (non-catalytic) and its thermos-stability and functional activity was enhanced as compared to wild type [25]. In another study, the calcium ligand binding site of endoglucanase was studied by replacing the amino acid residues and the thermos-stability and activity of endoglucanase was increased as compared wild type endoglucanase [26]. The thermos-stability and activity were enhanced by changing amino acid residues which were specifically involved in ligand binding under temperature modulation based upon A B approach analysis for endoglucanase B from *T. reesei* [27]. Bayram et al., (2015) also studied amino acids residues involved in thermos-stability and the mutated one showed better results as compared to wild type, both in RD and functional analysis studies [28]. The structural subdomain of endoglucanase B was mutated which were comprised of 56 amino acids and it yielded stable and effective enzyme function as compared to wild type [29]. The affirmative studies correlate to the conducted study, highlighting the regions involved in enzyme stability and activity other than active site residues.

In addition to recombinant DNA technology, researchers have also employed other techniques like solid state fermentation for production of cellulase enzymes to resolve the issue of use of lignocellulosic biomass. The worldwide research is still ongoing to find a better cellulase enzyme cocktail that can efficiently hydrolyze biomass and be used for other biotechnological applications [30,31]. Different *A. niger* strains have been exploited for cellulase production using conventional fermentation techniques [32]. However, we found that incorporation of mutagenesis to the fungal cultures followed by cellulase production through gene expression into suitable host is more successful tool to enhance the enzyme activity and bioinformatics tools can be applied to investigate the effect of mutagenesis on enzyme production.

## Conclusion

The mutant cellulase gene from an indigenous strain of *A. niger* was successfully expressed into *pET28a+* and cellulytic potential of mutant enzyme was found 4.5 times more than native enzyme using CMC as substrate, due to mutation of five nucleotide base pairs. The mutant enzyme was also found more thermostable at 50°C as compared to native at 35°C. The effect of metal ions on enzyme activity was also investigated and it was found that $Mg^{2+}$ and $Ca^{2+}$ maximally induced enzyme activity while it was decreased by $Cu^{2+}$, $Hg^{2+}$ and $Zn^{2+}$ ions. The integrity of recombinant mutant endoglucanase B from *A. niger* structure was explored through CHARMM36 force field, GROMACS, Parrinello-Rahman approach, PyMOL and VMD bioinformatics tools. A reduction in the root mean square deviation (RMSD) was found between protein and its ligand which served as a favourable indication of structural resemblance and stability. A low RMSF value suggested a region with greater rigidity. Lower value of Rg indicated a denser molecular arrangement. A low SASA of mutant than native endoglucanase B indicated a molecule that is compact and has a limited number of exposed surfaces. Hence, the produced recombinant mutant cellulase was found more cellulytic as compared to native enzyme and could be employed for further industrial investigations.

## Supporting information

**S1 Fig. 0.8% Agarose gel electrophoresis of extracted genomic RNA.** Lane M represents 1 Kb DNA ladder (Invitrogen). Lane 1 represents mRNA of native and Lane 2 represents mRNA of mutant strain of *A. Niger*.
(TIF)

**S2 Fig. Agarose gel electrophoresis of PCR amplified native cellulase gene from *A. niger*.** Lane M represents Gene Ruler 1 kb DNA ladder for size comparison of amplified gene and Lane 1 represents DNA band of native endoglucanase B (993bp).
(TIF)

**S3 Fig. Agarose gel electrophoresis of PCR amplified mutant cellulase gene from *A. niger*.** Lane M represents Thermo scientific Gene Ruler 1 kb DNA ladder for size comparison of amplified gene and Lane 1 represents DNA band of mutant endoglucanase B (993bp).
(TIF)

**S4 Fig. Protein sequence alignment of native (Query) and reported (O74706 · EGL-B_ASPNG) endoglucanase B.**
(TIF)

**S5 Fig. Protein sequence alignment of reported (O74706 · EGLB_ASPNG) (ENDoB) and mutant (muENDoB) endoglucanase B.**
(TIF)

**S6 Fig. Restriction double digestion of cloning vector showing 5375bp band of *pET*28a (+) vector and 993bp of endoglucanase B gene.**
(TIF)

**S7 Fig. SDS-PAGE analysis of fractions collected after Ni-affinity chromatography for mutant endoglucanase B purification.** M: Mobilities of proteins of known molecular masses on SDS-PAGE, Lane 1–6: Fractions loaded from 2nd to 3rd of 200–250 mM of imidazole respectively.
(TIF)

**S8 Fig. SDS-PAGE analysis of fractions collected after Ni-affinity chromatography for native endoglucanase B purification.** Lane M: Mobilities of proteins of known molecular masses on SDS-PAGE. Lane 1–3: showing purified dialysed fraction of endoglucanase B native.
(TIF)

**S1 Table. Forward and reverse primers used to clone cellulase gene.**
(TIF)

## Acknowledgments

The authors are thankful to Dr. Naeem Mehmood Ashraf and Saira Ahmad for their help in English proof reading of manuscript.

## Author Contributions

**Conceptualization:** Muddassar Zafar.

**Data curation:** Waqas Ahmad.

**Funding acquisition:** Muddassar Zafar.

**Investigation:** Zahid Anwar.

**Methodology:** Waqas Ahmad, Muddassar Zafar.

**Software:** Waqas Ahmad.

**Supervision:** Muddassar Zafar, Zahid Anwar.

**Validation:** Waqas Ahmad, Muddassar Zafar.

**Visualization:** Zahid Anwar.

**Writing – original draft:** Waqas Ahmad.

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
