## [Decision Letter · Decision Letter 0]

28 Feb 2024

PONE-D-24-03884Heterologous expression and characterization of mutant cellulase from indigenous strain of Aspergillus nigerPLOS ONE

Dear Dr. Zafar,

Thank you for submitting your manuscript to PLOS ONE. After careful consideration, we feel that it has merit but does not fully meet PLOS ONE’s publication criteria as it currently stands. Therefore, we invite you to submit a revised version of the manuscript that addresses the points raised during the review process.

We look forward to receiving your revised manuscript.

Kind regards,

Habibullah Nadeem

Academic Editor

PLOS ONE

Journal Requirements:

"The research work was funded by an NRPU Project No. 6485, granted by Higher Education Commission (HEC), Government of Pakistan."

"The research work was funded by an NRPU Project No. 6485, granted by Higher Education Commission (HEC), Government of Pakistan."

Please be informed that funding information should not appear in the Acknowledgments section or other areas of your manuscript. We will only publish funding information present in the Funding Statement section of the online submission form. 

"The research work was funded by an NRPU Project No. 6485, granted by Higher Education Commission (HEC), Government of Pakistan."

7. We are unable to open your Supporting Information file [Supporting Information Figures and Table.rar]. Please kindly revise as necessary and re-upload.

Reviewers' comments:

Reviewer's Responses to Questions

**Comments to the Author**

1. Is the manuscript technically sound, and do the data support the conclusions?

Reviewer #1: Yes

Reviewer #2: Yes

2. Has the statistical analysis been performed appropriately and rigorously? 

Reviewer #1: Yes

Reviewer #2: Yes

3. Have the authors made all data underlying the findings in their manuscript fully available?

Reviewer #1: Yes

Reviewer #2: Yes

4. Is the manuscript presented in an intelligible fashion and written in standard English?

Reviewer #1: Yes

Reviewer #2: Yes

5. Review Comments to the Author

Reviewer #1: However, we recommend that authors make some minor corrections before publication.

1. Rewrite the abstract: Purpose; Methods; Results and Conclusion

2. Increase the novelty of your introduction. What makes this work more promising than the the previous research in the literature.

3. In all over the manuscript, some recent relevant references about production and application should be included in appropriate place:

10.1007/s12155-020-10157-0

10.1007/s12649-016-9810-z

10.1007/s10068-013-0001-4

10.1590/S0103-84782011005000145

Regards

Reviewer #2: I have gone through the manuscript entitled, “Heterologous expression and characterization of mutant cellulase from indigenous strain of Aspergillus niger” and recommends the ACCEPTANCE of manuscript for the publication in PLOS One in view of following observations/minor corrections:

1. The manuscript describes an original research work elaborating expression of cellulase gene from an indigenous strain of Aspergillus niger and significantly contributes to the knowledge of microbial gene expressions for successful hyper expression of enzymes (Cellulase).

2. The experiments have been performed upto the mark and have been described in a detailed way.

3. Experimental conclusions are in accordance with the findings of the researchers.

4. Under sub heading; Solvent Accessible Surface Area (SASA), The statement describing stability of endoglucanase B is incomplete and it should be written in the past tense rather than present tense.

5. Same suggestion for the statement under sub heading; hydrogen bond analysis of wild and mutant type of endoglucanase B

6. On Page 28, there should be proper space among values of predicted ligand binding residues.

7. There is need to mention the version of Reprof software used for predicting wild type residues.

6. PLOS authors have the option to publish the peer review history of their article (what does this mean?). If published, this will include your full peer review and any attached files.

Reviewer #1: **Yes: **Marcelo Franco

Reviewer #2: No

---

## [Author Response · Author response to Decision Letter 0]

27 Mar 2024

RESPONSE TO REVIEWERS COMMENTS

Reviewer 1:

S. No. Comment Response

1 Rewrite the abstract: Purpose; Methods; Results and Conclusion The purpose, method, results and conclusion of the research work has already been mentioned very clearly in the abstract of the manuscript. If the reviewer wants to add these as sub-headings in the abstract, it will not be in accordance with the format of PLOS ONE journal. Therefore, sub-headings have not been added in the abstract.

2 Increase the novelty of your introduction. What makes this work more promising than the the previous research in the literature. The novelty/significance of research work statement has been incorporated into introduction chapter of the revised manuscript.

3 In all over the manuscript, some recent relevant references about production and application should be included in appropriate place: The relevant suggested references about the production and application have been incorporated into manuscript as per reviewer’s suggestion.

Reviewer 2:

S. No. Comment Response

1 Under sub heading; Solvent Accessible Surface Area (SASA), The statement describing stability of endoglucanase B is incomplete and it should be written in the past tense rather than present tense. The correction has been incorporated accordingly. 

2 Same suggestion for the statement under sub heading; hydrogen bond analysis of wild and mutant type of endoglucanase B The correction has been incorporated accordingly.

3 On Page 28, there should be proper space among values of predicted ligand binding residues. The correction has been incorporated accordingly.

4 There is need to mention the version of Reprof software used for predicting wild type residues. The name of version of software to predict the wild type residues has been mentioned.

---

## [Decision Letter · Decision Letter 1]

18 Apr 2024

Heterologous expression and characterization of mutant cellulase from indigenous strain of Aspergillus niger

PONE-D-24-03884R1

Dear Dr. Zafar,

We’re pleased to inform you that your manuscript has been judged scientifically suitable for publication and will be formally accepted for publication once it meets all outstanding technical requirements.

Kind regards,

Habibullah Nadeem

Academic Editor

PLOS ONE

Additional Editor Comments (optional):

Reviewers' comments:

Reviewer's Responses to Questions

**Comments to the Author**

1. If the authors have adequately addressed your comments raised in a previous round of review and you feel that this manuscript is now acceptable for publication, you may indicate that here to bypass the “Comments to the Author” section, enter your conflict of interest statement in the “Confidential to Editor” section, and submit your "Accept" recommendation.

Reviewer #1: All comments have been addressed

Reviewer #2: All comments have been addressed

2. Is the manuscript technically sound, and do the data support the conclusions?

Reviewer #1: Yes

Reviewer #2: Yes

3. Has the statistical analysis been performed appropriately and rigorously? 

Reviewer #1: Yes

Reviewer #2: Yes

4. Have the authors made all data underlying the findings in their manuscript fully available?

Reviewer #1: Yes

Reviewer #2: Yes

5. Is the manuscript presented in an intelligible fashion and written in standard English?

Reviewer #1: Yes

Reviewer #2: Yes

6. Review Comments to the Author

Reviewer #1: Dear

The authors revised the manuscript carefully and I recommend for publication in current format.

Reviewer #2: All the necessary comments raise by me has been incorporated by the author.Now consider this article for further process.

7. PLOS authors have the option to publish the peer review history of their article (what does this mean?). If published, this will include your full peer review and any attached files.

Reviewer #1: **Yes: **Marcelo Franco

Reviewer #2: No

---

## [Editor Report · Acceptance letter]

3 May 2024

PONE-D-24-03884R1 

PLOS ONE

Dear Dr. Zafar, 

I'm pleased to inform you that your manuscript has been deemed suitable for publication in PLOS ONE. Congratulations! Your manuscript is now being handed over to our production team.

Kind regards, 

on behalf of

Dr. Habibullah Nadeem 

Academic Editor

PLOS ONE